# Evaluation of Stability and Accuracy Compared to the Westergren Method of ESR Samples Analyzed at VES-MATIC 5

**DOI:** 10.3390/diagnostics14050557

**Published:** 2024-03-06

**Authors:** Maria Lorubbio, Daniela Diamanti, Alessandro Ghiandai, Carolina Pieroni, Donatella Bonini, Massimiliano Pettinari, Gabriele Gorini, Stefania Bassi, Paola Meloni, Agostino Ognibene

**Affiliations:** 1Chemical-Clinical Analysis Laboratory, Department of Laboratory Medicine and Transfusion, San Donato Hospital, 52100 Arezzo, Tuscany, Italy; a.ghiandai2@student.unisi.it (A.G.); donatella.bonini@uslsudest.toscana.it (D.B.); massimiliano.pettinari@uslsudest.toscana.it (M.P.); gabriele.gorini@uslsudest.toscana.it (G.G.); stefania.bassi@uslsudest.toscana.it (S.B.); paola.meloni@uslsudest.toscana.it (P.M.); agostino.ognibene@uslsudest.toscana.it (A.O.); 2DIESSE-Diagnostica Senese S.p.A., Strada dei Laghi, 35-39, 53035 Monteriggioni, Siena, Italy; danieladiamanti@diesse.it (D.D.); carolinapieroni@diesse.it (C.P.)

**Keywords:** ESR, stability, EDTA

## Abstract

The Erythrocyte Sedimentation Rate (ESR) is a diagnostic estimator of systemic inflammation as a reflection of acute phase proteins circulating in the blood. The purpose of this manuscript is to evaluate the blood stability at room temperature (RT) and at 4 °C to avoid ESR diagnostic errors, as well as the accuracy of the VES-MATIC 5 analyzer. The ESR stability evaluation at RT for 24 h (4 h “T1”, 6 h “T2”, 8 h “T3”, 10 h “T4”, 24 h “T5”) and at 4 °C (24 h, 36 h, 48 h) was carried out using 635 total samples, starting with T0 (2 h of venipuncture). For method comparison, 164 patients were analyzed using VES-MATIC 5 and then the Westergren reference method. The sample at RT is established by a significant gradual decrease in correlation R = 0.99 (T0 vs. T1), R = 0.97 (T0 vs. T2), R = 0.92 (T0 vs. T3), R = 0.87 (T0 vs. T4), and R = 0.40 (T0 vs. T5). The stability at 4 °C after 24 h, 36 h, and 48 h showed a regression of R = 0.99, R = 0.97, and R = 0.95, respectively. Therefore, ESR measurements on RT samples beyond 6 h after collection cannot be carried out, but the ESR can be measured until 36 h for samples stored at 4 °C. Moreover, the VES-MATIC 5 accuracy performance compared to the Westergren method (R = 0.96) is confirmed.

## 1. Introduction

The Erythrocyte Sedimentation Rate (ESR) measures the rate (mm/h) at which red blood cells form aggregates (or rouleaux) [1], and it does not determine an analyte but rather a physical phenomenon [2]. The sedimentation curve is sigmoidal and comprises three phases: the lag phase, sedimentation phase, and packing phase [3]. In the first phase (lag phase), erythrocytes dispersed in plasma form one-dimensional coin stacks (rouleaux). Rouleaux form aggregates by gathering in two- to three-dimensions over time, and sedimentation of the erythrocyte–plasma interface occurs after a certain delay. At this time, the aggregate size increases according to the plasma concentration of fibrinogen or globulin [1,4] and decreases as the hematocrit increases [5,6,7,8]. The main phase of the ESR is the second (sedimentation) phase, in which the sedimentation rate becomes maximum and almost constant. In the third (packing) phase, the sedimentation rate is reduced by the deposition of erythrocytes at the bottom of the tube. Finally, the sedimentation distance converges on the value corresponding to the volume ratio of blood cells and plasma over time [3]. The ESR is an estimator of the severity of systemic inflammation [9] because it depends on the concentration of acute-phase proteins circulating in the blood, particularly fibrinogen; these proteins increasing the dielectric constant in the blood neutralize the negative charges on the surface of red blood cells, which repel one another and physiologically oppose aggregation [2]. The ESR clinical contribution is not specific for any single disease but is used in combination with other tests to determine the presence of increased inflammatory activity. The ESR has long been used as a “sickness indicator” due to its reproducibility and low cost [10]. Despite its limitations of low sensitivity and specificity compared to more specific inflammatory markers, ESR is still widely employed for the diagnosis and monitoring of a variety of pathological conditions, particularly infections and rheumatic diseases [11], due to its simplicity, speed availability, and low cost of the test [5]. Moreover, elevations in ESR are associated with infarctions, autoimmune diseases, and malignancies [4,6]. High ESR levels have been notably linked to poor prognosis in various cancers, including hematological diseases like Hodgkin’s lymphoma [12].

Moreover, in a recent study, ESR was shown to be a strong predictor of mortality from coronary heart disease and appears to be a marker for aggressive forms of the disease [13,14]. An elevated ESR may be an important adjunct in detecting coronary artery disease. This is possibly linked to the inflammatory condition of coronary disease [15,16,17]. Similarly, there may exist a relationship between the ESR in ischemic stroke and the amount of local brain injury, atherosclerosis, and short-term outcomes. It also responds to surgical intervention [10].

Thus, ESR has a diagnostic value in some conditions and allows for the monitoring of therapeutic interventions in others [7,8].

ESR is also affected by red blood cell (RBC)-related factors (size, shape, number, surface charge, and aggregation of erythrocytes) and hematological parameters like hematocrit and hemoglobin. Therefore, the ESR phenomenon is determined not only from plasma protein concentrations but also in patients with anemia or high red blood cell volume where fibrinogen [16] tends to be higher [2].

Indeed, although many inflammatory illnesses will increase the ESR, other conditions exist that can lower the ESR. These “lowering factors” can exist either as isolated disease processes or in conjunction with other pathologic conditions that raise the ESR, thus giving a “lower than expected” ESR result in light of a serious underlying inflammatory process [18]. 

Polymyalgia rheumatica and giant cell (temporal) arteritis are inflammatory conditions that usually occur in persons older than 50 years [18]. In rheumatoid arthritis, systemic lupus erythematosus, and osteoarthritis, ESR levels are frequently used to help make the diagnosis. An elevated ESR may be useful in the diagnosis and follow-up in patients with osteomyelitis. Optimal ESR cutoff levels for the diagnosis are variable. In cases of proven osteomyelitis, the ESR may be used to monitor responses to therapy or relapse [10]. ESR may be used as an indicator of the presence of invasive bacterial infection in children. Polycythemia (an increased number of red blood cells) increases blood viscosity and can cause a reduced ESR (reduces the rate at which RBC rouleaux will settle to the bottom of the Westergren tube) [19]. Some hemoglobinopathies, such as sickle cell disease or Spherocytosis (the presence of sphere-shaped rather than disk-shaped RBCs) inhibit rouleaux formation and can decrease the ESR [20].

A normal ESR has a high negative predictive value for these conditions [5,11,21]. Pregnancy and aging may also increase the ESR. Any process that elevates fibrinogen levels (e.g., pregnancy, infection, diabetes mellitus, end-stage renal disease, heart disease, malignancy) may also elevate the ESR [8]. Measurements of CRP concentration and ESR are frequently ordered jointly in clinical practice [22]; the majority of previous studies that have investigated the agreement between CRP and ESR values have included selected samples of patients with specific inflammatory diseases [2]. Determining the ESR in general populations is important for interpreting reference values. The guidelines for the definition and determination of reference intervals indicate that partitioning should be considered when there are significant differences among subgroups defined by age, sex, and common exposures [5,21]. The ESR increases with age in adults, and at a given age is higher in females than in males [5,11,22,23,24]. 

The reference method for ESR measurement [4] is the Westergren method, which constitutes the gold standard as recommended by the International Council for Standardization in Hematology (ICSH) [7,11]. The Westergren method uses a whole blood sample diluted with sodium citrate anticoagulant (4:1), and the value of ESR is determined after one hour in a vertically placed tube [8]. In 1993, ICSH described a standardized method as an alternative to, and potential replacement for, the reference method. For working (routine) methods, ICSH recommended specifications for selected methods. They were procedures whose reliability was verified against the reference or standardized method as undilute sampled, and which minimized the biohazard risk of the test procedure [25].

In 2017, the ICSH group published new recommendations, stressing the importance of ensuring that measurements obtained in different laboratories were comparable [7]. ICSH and CLSI therefore made new recommendations in 2010 and 2011 [23,26]. The ICSH document recognized that automated methods were routinely used in many laboratories, using diluted or undiluted samples. The reference procedure remained based on the Westergren method. The document stated that all new technologies, instruments, or methodologies had to be evaluated against the Westergren reference method before being introduced into clinical use and that “systems that give the results as the Westergren method with diluted blood at 60 min or normalized to 60 min are the only ones of clinical value”. It was recommended that manufacturers provide data on the reliability and trueness of any method and instrument, as well as calibration and control procedures.

The ICSH has recently proposed a classification of commercialized ESR methods [7] proposed by the companies to shorten the time of the ESR measurement [22] and increase the laboratory turnaround time (TAT). In addition to the gold standard, two other categories of ESR methods are defined: the modified Westergren method, which is based on Westergren methodology with acceptable modifications, including shorter analysis times and the use of anticoagulants other than citrate, and the alternative method incorporating new technologies for ESR measurement [7,23]. The adoption of these semi-automatic and fully automated systems brings numerous advantages, in addition to the reduction of the analysis times, such as a decrease in the costs of the sampling devices and the blood volume necessary for the test. Moreover, the use of ethylenediaminetetraacetic acid (EDTA) anticoagulant blood improves sample stability [24], and the introduction of analyzers running these samples makes it possible to employ the whole blood sample withdrawn for other hematology tests [22], as well as for ESR measurement, providing patient and operator safety, reducing sample handling, and streamlining workflow [23]. The stability evaluation is a crucial aspect of the preanalytical phase to guarantee the diagnostic quality of laboratory results [20]. Little research has been conducted on the stability of the EDTA whole blood sample at variable times and temperatures for the ESR test [27,28]. The purpose of this work is to assess both the stability of EDTA samples for ESR analysis at room temperature for 24 h (4 h, 6 h, 8 h, 10 h, 24 h) and at 4 °C (24 h, 36 h, and 48 h), and the accuracy with respect to the Westergren method using the VES-MATIC 5 instrument (DIESSE Monteriggioni, Siena).

## 2. Materials and Methods

### 2.1. Samples

Random whole blood samples of the daily routine of the San Donato Hospital in Arezzo, collected in 3.0 mL Vacutainer K2-EDTA tubes (Becton Dickinson, London, UK) for CBC (Complete Blood Count) from May to July 2023, were used for this study. This study was approved by a local ethical committee and was conducted following the requirements of the Declaration of Helsinki.

### 2.2. ESR Stability Study Design

The evaluation of stability at room temperature for 24 h (4 h, 6 h, 8 h, 10 h, 24 h) and at 4 °C (24 h, 36 h, 48 h) ESR was measured within 2 h of venipuncture, and for fresh samples, measurements were indicated as T0 (day 1).

*Stability at room temperature for 24 h.* ESR was evaluated at T0 and every 2 h from the previous read, storing samples at room temperature (20–26 °C). Time points collected during day 1 were T1 after 4 h from collection, T2 (at 6 h), T3 (at 8 h), T4 (at 10 h), and T5 (at 24 h).

*Stability at 4 °C for 24 h.* After the analysis at T0, blood samples were stored at 4–6 °C for 24 h. On day 2, tubes were stored at room temperature for 1 h, gently shaken 10–12 times every 20 min, and loaded into VES-MATIC 5 for ESR analysis (T1 at day 2). After 3 h, the test was retried to also have these detections.

*Stability at 4 °C for 36 h.* ESR was measured at T0 and samples were stored at 4–6 °C for 36 h. After storing tubes at room temperature for 1 h, mixing them every 20 min, samples were analyzed (T2 corresponding to 36 h of refrigeration).

*Stability at 4 °C for 48 h.* After T0 analysis, blood samples were stored at 4–6 °C for 48 h. On day 3, after allowing samples to acclimate for 1 h at room temperature and while shaking them, ESR was evaluated (T3 is 48 h of refrigeration).

## 3. Method Comparison

Samples were analyzed utilizing VES-MATIC 5 and then diluted with sodium citrate for the Westergren method within 4 h from collection.

### 3.1. The Westergren Method

The manual method of Westergren was performed according to the ICSH’s recommendations, diluting EDTA blood samples 4:1 with sodium citrate 3.8% directly into the test tube (FL-MEDICAL Torreglhia, Padua, Italy).

Particularly, mixing the blood with EDTA anticoagulant (15 mg/mL blood) is necessary at the time of venipuncture, but further mixing immediately before the ESR test is set up is critically important for reproducibility. For standard tubes (10–12 mm × 75 mm containing 5 mL blood and with an air bubble comprising at least 20% of the tube volume), there should be a minimum of eight complete inversions (180° × 2), with the air bubble traveling from end to end in the tube. Mixing must not cause hemolysis.

After gently mixing, samples were immediately aspirated into a graduated glass Westergren tube with a piston (FL-MEDICAL Torreglia, Padua, Italy). The tube was allowed to stand in a vertical position in the appropriate supporting rack, away from the sunlight. All the measurements stated at room temperature were carried out in the range of 20–22 °C. After 60 min, the ESR was recorded by visual inspection, reporting that the plasma column formed from the top of the tube to the RBC sedimentation level.

### 3.2. VES-MATIC 5 Analyzer Description

ESR measurement by VES-MATIC 5 system is based on the modified Westergren method, using the same EDTA-anticoagulated blood sample withdrawn for cell count hematological analyzer and loading the same tube rack.

After automatic sample mixing, the instrument detects the blood height in the tube and repeats the scanning after 12, 17, and 20 min by optoelectronic light sources. The differences between the first and the last record are transformed to 60 min Westergren (given in millimeter/hour) and corrected at 18 °C with the application of the Manley temperature algorithm. For each sample, the kinetics of sedimentation are visualized. All levels of sedimentation are read in the closed tube, so there is no waste production and safety risk for operators.

### 3.3. Statistical Analysis

The Spearman’s rank correlation test was used to evaluate the agreement between the reference method and the analyzer VES-MATIC 5, and through the Passing–Bablok plot analysis. Passing–Bablok regression is a method for nonparametric regression analysis suitable for method comparison studies, introduced by Wolfgang Bablok and Heinrich Passing in 1983. The procedure is adapted to fit linear errors-in-variables models. It is symmetrical and robust in the presence of one or a few outliers [29].

The bias and limits of agreement were estimated using a Bland–Altman plot. Bland–Altman plots are extensively used to evaluate the agreement between two different instruments or two measurement techniques, allowing for the identification of any systematic difference between the measurements or possible outliers. The mean difference is the estimated bias, and the SD of the differences measures the random fluctuations around this mean. If the mean value of the difference differs significantly from 0 on the basis of a one-sample t-test, this indicates the presence of a fixed bias. If there is a consistent bias, it can be adjusted for by subtracting the mean difference from the new method [30]. The normal distribution was assessed using a Kolmogorov–Smirnov test.

For the statistical analysis, ESR values > 140 mm/h obtained from VES-MATIC 5 were considered to be 140 mm/h. All statistical analyses were elaborated using GraphPad Prism statistical software (v. 2.01) and Microsoft Excel 365 (Microsoft Corporation, Redmond, WA, USA).

## 4. Results

The evaluation of stability was carried out using 635 total samples covering the entire analytical range divided into three ESR levels, low (<20 mm/h), medium (between 20 and 60 mm/h), and high (between 60 and 140 mm/h), proportionally represented.

The evaluation of the ESR test stability at room temperature in 155 (n.66 low; n.34 middle; n.55 high) samples was carried out for 24 h (4 h “T1”, 6 h “T2”, 8 h “T3”, 10 h “T4”, 24 h “T5”) starting from T0. Figure 1 shows the stability of the samples stored at room temperature. The Passing–Bablok plots showed a gradual decrease in correlation R = 0.99 from the comparison between T0 and T1 (Figure 1a), R = 0.97 between T0 and T2 (Figure 1c), R = 0.92 between T0 and T3 (Figure 1e), R = 0.87 between T0 and T4 (Figure 1g), and, finally, R = 0.40 between T0 and T5 (Figure 1i). The results obtained from the samples utilized for all evaluations are representative of the entire analytic range of the studied instrument (1–140 mm/h).

The Bland–Altman test expresses between T0 and T1, bias = −0.65 (+1.96 SD = 10.77, −1.96 SD = −12.06) in Figure 1b; between T0 and T2 bias = −2.97 (+1.96 SD = 20.46, −1.96 SD = −26.39) in Figure 1d; between T0 and T3 bias = −6.82 (+1.96 SD = 29.45, −1.96 SD = −43.09) in Figure 1f; between T0 and T4 bias = −11.42 (+1.96 SD = 32.74, −1.96 SD = −55.58) in Figure 1h; and between T0 and T5 bias = −39.42 (+1.96 SD = 42.49, −1.96 SD = −121.33) in Figure 1. Figure 2 represents the distribution of the sample population used to evaluate stability at room temperature through the box plots of the samples at different times of sampling T0 mm/h (25° percentiles = 11.0, median = 33.0, 75° percentiles = 85.0), T1 at 4 h (25° percentiles = 8.0, median = 35.5, 75° percentiles = 84.5), T2 at 6 h (25° percentiles = 8.5, median = 26.0, 75° percentiles = 82.5), T3 at 8 h (25° percentiles = 7.0, median = 22.0, 75° percentiles = 81.0), T4 at 10 h (25° percentiles = 6.0, median = 17.5, 75° percentiles = 66.5), and T5 at 24 h (25° percentiles = 1.0, median = 2.0, 75° percentiles = 6.5). The differences in the reduction of the ESR value due to the decay of the samples stored at room temperature are always statistically significant when comparing T0 with T1 (*p* < 0.05), T2, T3, T4, and T5 (*p* < 0.0001).

Figure 3 shows the stability of the samples stored at 4 °C and analyzed at T1 after 24 h (146 samples: n.70 low; n.39 middle; n.37 high), T2 after 36 h (170 samples: n.76 low; n.43 middle; n.51 high), and T3 after 48 h (164 samples: n.85 low; n.34 middle; n.45 high) from T0 (2 h after collection). The Passing–Bablok regression shows a decrease in correlation from T1 to T2 to T3 compared to each T0 of R = 0.99, R = 0.97, and R = 0.95, respectively (Figure 3a–c). The Bland–Altman test shows that T1 bias = −2.28 (+1.96 SD = 10.99, −1.96 SD = −15.55), T2 bias = −5.37 (+1.96 SD = 15.23, −1.96 SD = −25.97), and T3 bias = −8.11 (+1.96 SD = 16.51, −1.96 SD = −32.73) compared to the respective T0 (Figure 3d–f). The distributions of sample populations to evaluate stability at 4 °C are depicted in Figure 4 at 24 h (a), 36 h (b), and 48 h (c). The box plots represent T0 mm/h (25° = 11.0, median = 22.0, 75° = 60.5) of the 24 h mm/h (25° = 8.5, median = 18.0, 75° = 58.5), T0 mm/h (25 ° = 10.0, median = 24.0, 75° = 73.0) of the 36 h mm/h (25° = 8.0, median = 20.0, 75° = 65.0), and T0 mm/h (25° = 8.0, median = 18.0, 75° = 68.5) of the 48 h mm/h (25° = 6.0, median = 13.0, 75° = 55.0). The differences due to the decay of the samples stored at 4 °C measured at T0 and after 24 h and 36 h is statistically not significant (*p* = NS), while the comparison of the results between T0 and 48 h stored samples showed statistically significant differences (*p* < 0.05). For the comparison of methods, 164 patients were enrolled (86 women and 78 men; mean age 67 ± 15.9 SD years). Figure 5 represents the comparison of methods between VES-MATIC 5 and Westergren’s manual one. The Passing–Bablok analysis shows a correlation of R = 0.96 (Figure 5a); the Bland–Altman test presents bias = 2.01 (+1.96 SD = 22.83, −1.96 SD = −18.82) in Figure 5b.

## 5. Discussion

The ICSH recommendations of 2017 defined in the classification of the methods to be used for the ESR measurement, in addition to the gold standard, also modified the Westergren methods and alternate ESR methods, which allow for the use of the whole blood sample in EDTA anticoagulant with a reduction in analysis times and risks for the patient and the operator performing the test.

The advantages of integration of ESR technology into automated systems include savings on labor, no need for aliquots (and therefore more efficient use of sample volumes), shorter turnaround times, and minimal exposure of laboratory staff to biohazards. The disadvantages include possible higher costs of instrumentation. Most publications compared the new instruments to the Westergren method. Some of these papers were not fully conclusive, stressing the importance of careful study designs [7].

It is of fundamental importance to know the stability of the samples for ESR analysis based on the adopted method, especially for daily practice, as a sample withdrawn from a few days earlier and poorly preserved must not be accepted to perform the test to avoid a diagnostic error. According to the results found in this work, the ESR value already begins to decay at room temperature in a statistically significant way 6 h after collection, so the analysis of specimens stored at room temperature already 6 h after collection cannot be performed, as observed by Hu et al. [28], who used samples diluted in sodium citrate, as well as by other authors [26,31]. This result can be seen clearly from the decrease in the correlation coefficient R, which is 0.99 at 4 h after sampling and progressively decreases to 0.40 after 24 h (Figure 1). The blood deterioration can also be seen from the similar decline in the medians calculated from the box plots of the distributions of the populations at the different times starting from median = 33.0 mm/h at T0 up to median = 17.5 mm/h of T4 (Figure 2). The reason for this variation in time is unknown, but it may, in part, be accounted for by differences in disease pathology [21]. ESR stability degrades over time due to changes in the shape of the red blood cells, which become spherical. These spherical cells are difficult to aggregate, affecting rouleaux formation. In addition, the interaction of charges between the membrane surface of the red blood cells and the plasma proteins is modified [28]. The results are certainly different for the samples refrigerated at 4 °C, in which there is no statistically significant decay at 24 h and 36 h compared to each T0 (2 h from sampling); the deterioration of the samples, however, is significant 48 h from T0. Therefore, blood samples, even if stored at 4 °C, cannot be analyzed after 2 days. The results of this work on the stability of samples stored at 4 °C agree with what was reported by other authors [26,27,28,29,30,31,32], who used samples in EDTA, but not with Hu, who used samples diluted in sodium citrate, finding that the samples are stable at 4 °C for up to 12 h. This consideration highlights the greater stability of erythrocyte morphology when using EDTA samples compared to those diluted in sodium citrate, contributing to the standardization of ESR determination [18]. Moreover, this study investigated for the first time the ESR stability at 4 °C for 36 h and suggests the possibility of test execution up to this time of refrigeration. Finally, this contribution evaluated the VES-MATIC 5 instrumental performances in terms of accuracy compared to the Westergren reference manual method, with a good correlation on the Passing–Bablok regression of R = 0.96 confirming the same result as Piva et al. [19]. The accuracy of the VES-MATIC 5 instrument is better than that of the VES-MATIC CUBE 200 instrument, which is based on the same method as calculated by different authors [22,31], but is also better than instruments with an alternative method [18].

## Figures and Tables

**Figure 1 diagnostics-14-00557-f001:**
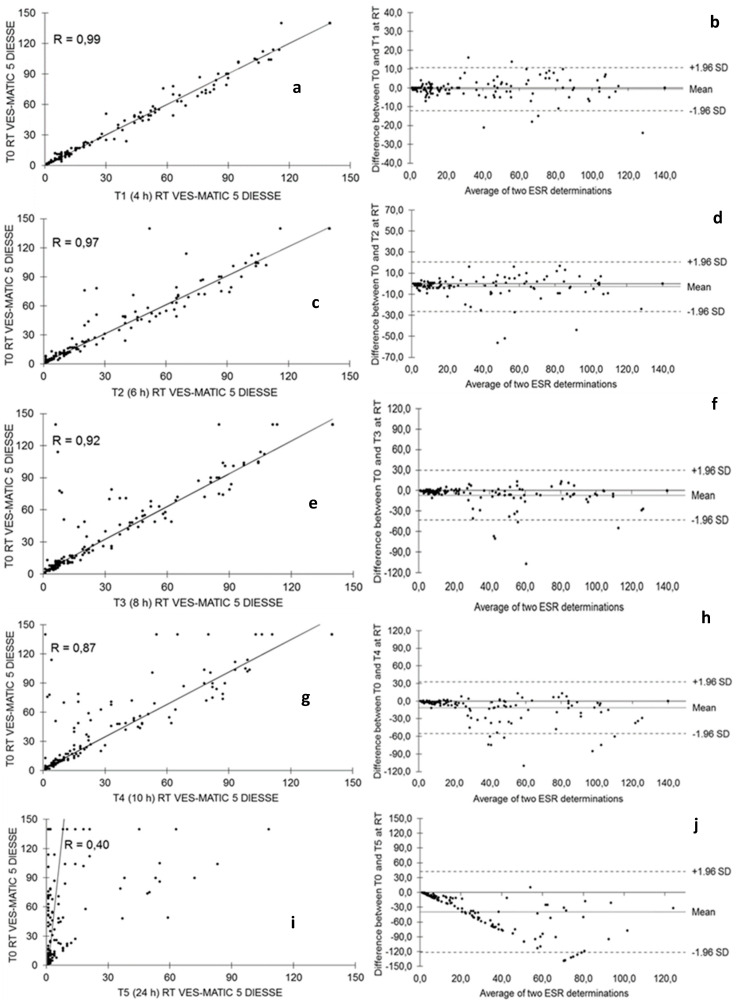
(**a**–**j**) Evaluation of stability for ESR test in blood samples stored at room temperature for 4, 6, 8, 10, and 24 h with Passing–Bablok and Bland–Altman tests.

**Figure 2 diagnostics-14-00557-f002:**
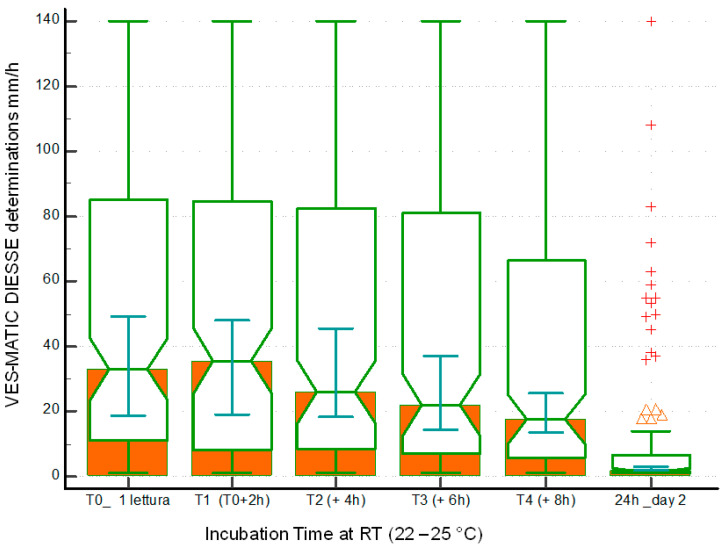
This graph shows the incubation times from T0 to T5 of the same sample studied, which is represented with a single notched box and whiskers plot. It is interesting to note the decay of the ESR values as a function of time; triangle: upper adjacent value, +: far out value.

**Figure 3 diagnostics-14-00557-f003:**
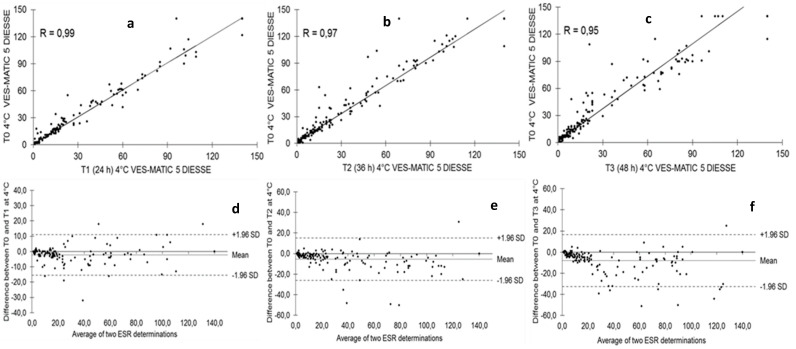
(**a**–**f**) Evaluation of stability for blood samples stored at 4 °C and analyzed after 24, 36, and 48 h from T0.

**Figure 4 diagnostics-14-00557-f004:**
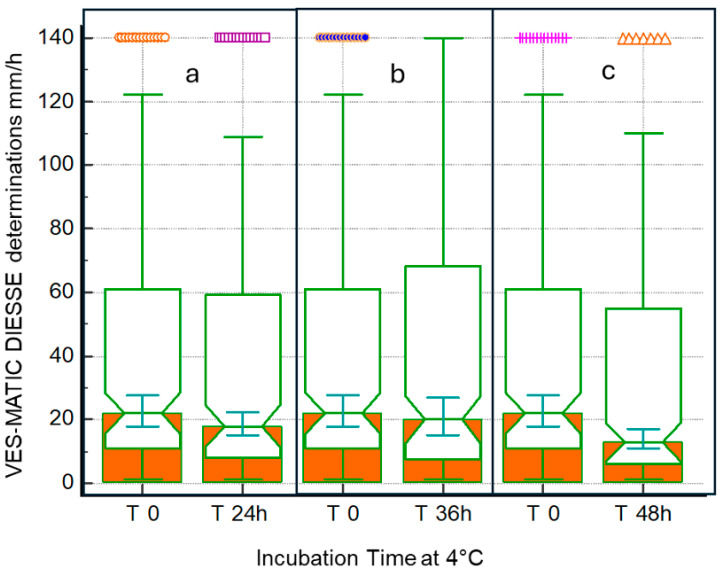
This notched box and whiskers plot shows three populations of samples at different incubation times from T0 to T24h, T36h, and T48h, respectively. In boxes a, b, and c, it is possible to note the variation in median value; the symbols at the top of each notched box and whiskers: far out values.

**Figure 5 diagnostics-14-00557-f005:**
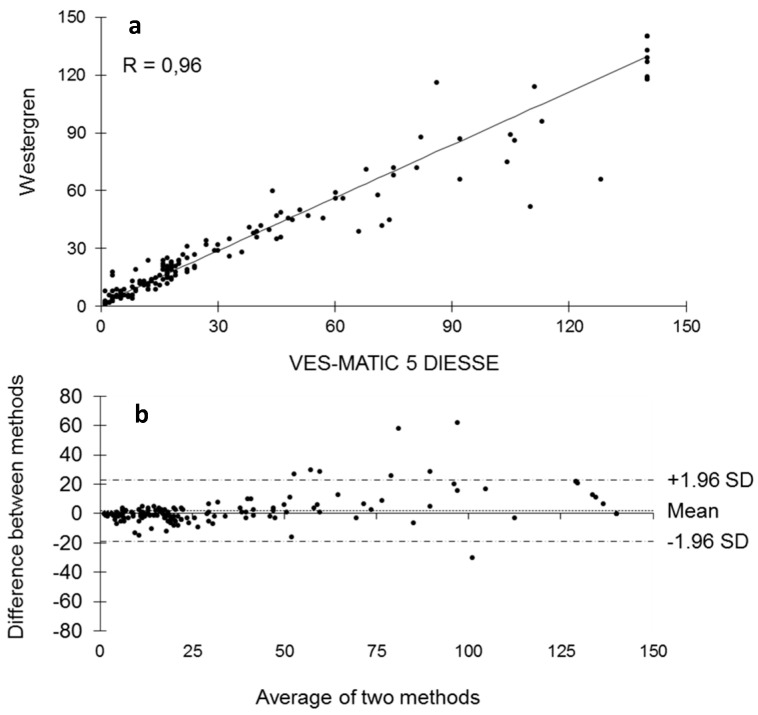
Passing–Bablok regression and Bland–Altman analysis between the measurement of ESR with the Westergren reference method and the VES-MATIC 5.

## Data Availability

The dataset used during this study is available from the corresponding authors upon reasonable request.

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
