# Peer review of "Evaluation of Stability and Accuracy Compared to the Westergren Method of ESR Samples Analyzed at VES-MATIC 5"

_diagnostics, 2024, doi:10.3390/diagnostics14050557_

Round 1

Reviewer 1 Report

Comments and Suggestions for Authors

Dear Authors

In Your manuscript "Evaluation of Stability and Accuracy Compared to the 2 Westergren Method of ESR Samples Analyzed at VES-MATIC 5" the results you are reporting are interesting.

Author Response

We thank the reviewer for positively evaluating the contribution of the manuscript.

Reviewer 2 Report

Comments and Suggestions for Authors

In this manuscript the Authors verified the accuracy of ESR measure of samples at room temperature for4, 6, 8, 10, 24 hours and at 4°C for 24, 36, 48 hours. The topic is quite interesting from a clinical practce point of view. The title is clear as well as the aim; results support the conclusions.

COMMENTS

L56 RBc --> red blood cells

L133 did you set the sample size before the enrollment. Otherwise move the number of blood sample etc in results. How did you assess the normal distribution?

L212-251 summarize the results as many findings are clearly reported in figures. Where is Fig1?

Fig 2 and 4 show the notched box and whiskers plot in order to make the readers understand if the sample size is sufficient

L275-282 is redundant. In general, you have to stress the originality of this study and its limitation. It is useless to repeat what is already reported in Methods. Please, rewrite the conclusions in order to make them clearer for the casual reader and stay more focused on the aim.

Author Response

Replay to the comments:

L56 RBc --> red blood cells.
The sentence has been corrected.

L133 set the sample size before registering. Otherwise move the blood sample number etc. in the results. How did you evaluate the normal distribution?
Thanks for the suggestions, the Material and the Method have been modified following your advice, in particular the number of samples studied has been reported in the results session.

L212-251 summarizes the results as many results are clearly reported in the figures. Where is Fig1?
Figure 1 was included in the manuscript.

Fig. 2 and 4 show the notched box and whisker diagram to help readers understand whether the sample size is sufficient.
The legend of figures 2 and 4 has been corrected to explain the phenomenon and make it clearer to readers. The sample size was declared in the results.

L275-282 is redundant. In general, it is necessary to underline the originality of this study and its limitations. It is useless to repeat what has already been reported in Methods. Please rewrite the conclusions to make them clearer for the casual reader and stay more focused on the goal.
The redundant part has been eliminated.

Reviewer 3 Report

Comments and Suggestions for Authors

The article evaluates the stability and accuracy compared to the Westergren method of ESR samples analyzed at VES-MATIC 5. The introduction is ample.The metodology is accurate. The importance of the blood analysis is described well. The resulta are clearly expressed. The disucssions are well conducted. Please discuss the importance of good performance of devices used in the hospital. Please describe the role of ESR in inflaamtory diseases . Please consult: Endres L, Tit DM, Bungau S, Pascalau NA, Èšodan LM, Bimbo-Szuhai E, Iancu GM, Negrut N. Incidence and Clinical Implications of Autoimmune Thyroiditis in the Development of Acne in Young Patients. Diagnostics (Basel). 2021 Apr 28;11(5):794. doi: 10.3390/diagnostics11050794. PMID: 33924808; PMCID: PMC8145646.

Comments on the Quality of English Language

English is ok.

Author Response

Replay to the rewiever comments

I am sorry to disagree with the reviewer's advice as the inflammatory process linked to autoimmune thyroiditis and the reported manuscript represents one of the many examples where the ESR is modified, in our present contribution we do not address the diagnostic function of the ESR but the effects of poor conservation of the sample. However, we are about to publish another manuscript on ESR where we will deal with diagnostic aspects of inflammation and we will certainly be able to insert this particular pathological profile in that context.

Round 2

Reviewer 2 Report

Comments and Suggestions for Authors

In Fig 2 and 3 there are not notched box and whiskers plots!

Author Response

Dear Reviewer
We try to answer at your comment.
Best regards

Answer to the comment "In Fig 2 and 3 there are not notched box and whiskers plots!"

The box and whiskers plots are in figure 2 and 4 and it'snt in the plot  in figure 3. In figure 2 and 4 has been corrected the legends about the kind of plot utilized. The corrections has been made following the present bibliography: 
- C., Dutoit, S. H. (2012). Graphical exploratory data analysis. Springer. ISBN 978-1-4612-9371-2. 
- Marmolejo-Ramos, F.; Tian, S. (2010). "The shifting boxplot. A boxplot based on essential summary statistics around the mean". International Journal of Psychological Research. 3 (1): 37–46. doi:10.21500/20112084.823.

Round 3

Reviewer 2 Report

Comments and Suggestions for Authors

In Fig 2 and 4 there are not notched box and whiskers plots! If you do not know what they are,  see https://sites.google.com/site/davidsstatistics/davids-statistics/notched-box-plots  or https://www.jstor.org/stable/2683468

Author Response

Dear reviewer
We are grateful for the advice, unfortunately the problem was not related to the knowledge of the type of graph but rather to the software to be used to show, the data already declared in the results, in the graph according to your wishes. Well, now we have obtained the appropriate software and inserted the figure 2 and 4 as  the "notched"  box and whiskers into the manuscript .
Best regards

Round 4

Reviewer 2 Report

Comments and Suggestions for Authors

All comments were addressed